# Upregulation of Heat-Shock Protein (hsp)-27 in a Patient with Heterozygous SPG11 c.1951C>T and SYNJ1 c.2614G>T Mutations Causing Clinical Spastic Paraplegia

**DOI:** 10.3390/genes14071320

**Published:** 2023-06-23

**Authors:** Juan Antonio García-Carmona, Joaquín Amores-Iniesta, José Soler-Usero, María Cerdán-Sánchez, Javier Navarro-Zaragoza, María López-López, Juan José Soria-Torrecillas, Ainhoa Ballesteros-Arenas, José Antonio Pérez-Vicente, Pilar Almela

**Affiliations:** 1Department of Neurology, Santa Lucia University Hospital, 30202 Cartagena, Spain; 2Group of Clinical & Experimental Pharmacology, Institute for Biomedical Research of Murcia (IMIB), 30120 Murcia, Spain; 3Department of Animal Health, University of Murcia, 30100 Murcia, Spain; 4Group of Mycoplasmosis, Epidemiology and Pathogen-Host Interaction, Institute for Biomedical Research of Murcia (IMIB), 30120 Murcia, Spain; 5Department of Biology and Biochemistry, University of Castilla-León, 09001 Burgos, Spain; 6Department of Pharmacology, University of Murcia, 30100 Murcia, Spain

**Keywords:** spastic paraplegia, spatacsin, synaptojanin, heat-shock protein 27, dopamine oxidation

## Abstract

We report a 49-year-old patient suffering from spastic paraplegia with a novel heterozygous mutation and analyzed the levels of heat shock proteins (hsp)-27, dopamine (DA), and its metabolites in their cerebrospinal fluid (CSF). The hsp27 protein concentration in the patient’s CSF was assayed by an ELISA kit, while DA levels and its metabolites in the CSF, 3,4-dihydroxyphenylacetic acid (DOPAC), Cys-DA, and Cys-DOPA were measured by HPLC. Whole exome sequencing demonstrated SPG-11 c.1951C>T and novel SYNJ1 c.2614G>T mutations, both heterozygous recessive. The patient’s DA and DOPAC levels in their CSF were significantly decreased (53.0 ± 6.92 and 473.3 ± 72.19, *p* < 0.05, respectively) while no differences were found in their Cys-DA. Nonetheless, Cys-DA/DOPAC ratio (0.213 ± 0.024, *p* < 0.05) and hsp27 levels (1073.0 ± 136.4, *p* < 0.05) were significantly higher. To the best of our knowledge, the c.2614G>T SYNJ1 mutation has not been previously reported. Our patient does not produce fully functional spatacsin and synaptojanin-1 proteins. In this line, our results showed decreased DA and DOPAC levels in the patient’s CSF, indicating loss of DAergic neurons. Many factors have been described as being responsible for the increased cys-DA/DOPAC ratio, such as MAO inhibition and decreased antioxidant activity in DAergic neurons which would increase catecholquinones and consequently cysteinyl-catechols. In conclusion, haploinsufficiency of spatacsin and synaptojanin-1 proteins might be the underlying cause of neurodegeneration produced by protein trafficking defects, DA vesicle trafficking/recycling processes, autophagy dysfunction, and cell death leading to hsp27 upregulation as a cellular mechanism of protection and/or to balance impaired protein trafficking.

## 1. Introduction

Hereditary spastic paraplegia is a rare syndrome comprising a group of neurodegenerative diseases that are genetically heterogeneous. Spastic paraplegia is caused by distal degeneration of lateral corticospinal tract axons causing a typical phenotype consisting of progressive and bilateral spasticity and weakness of the lower limbs [1,2]. The prevalence of hereditary spastic paraplegia is 1–10:100,000 in the US and Europe [3]. Non-complicated hereditary spastic paraplegia forms are characterized by other clinical features such as bladder dysfunction, hyperreflexia, gait impairment, and vibratory sense impairment [4]. Additionally, patients with complicated hereditary spastic paraplegia forms may suffer from skeletal or skin abnormalities, seizures or epilepsy, retinopathy, and Parkinsonism [5]. Genetic inheritance is known to have autosomal dominant (AD), autosomal recessive (AR), and X-linked forms [6]. Nonetheless, although a large number of phenotypes have been described and 82 causative genes have been identified, the molecular diagnosis rate is 30–60% [7,8]. Therefore, recent studies suggest that more unknown genes should be involved in spastic paraplegia [9]. It is therefore hardly surprising that the latest evidence in this field highlights genetic heterogeneity, the discovery that different mutations in a single gene and allelic heterogeneity may cause spastic paraplegia’s clinical features, resulting in complex gene interactions [10]. Furthermore, it has been demonstrated that the activity of many different proteins encoded by spastic-paraplegia-related genes converges from a few distinct pathophysiological mechanisms also regulated by other genes [10,11]. In this regard, recent studies have suggested that dysfunction in intracellular trafficking (lipid metabolism, the lysosome system, or organelle shaping) may lead to retrograde axonal degeneration of corticospinal, spinocerebellar, and sensory axons of the central nervous system [12,13,14]. Among other proteins, spatacsin (gen SPG11) and synaptojanin-1 (gen SYNJ1) have been described in the regulation of vesicle endocytosis and intracellular trafficking [15,16,17].

Small heat-shock proteins (sHsps) are a family of chaperone proteins [18]. Chaperones are known to interact with and stabilize misfolded and unfolded proteins, preventing their aggregation but also because they selectively target proteins and directly transport them into the lysosome for autophagy [18]. This process, known as chaperone-mediated autophagy (CMA), is essential to maintain intracellular proteostasis and avoid proteotoxicity in dopaminergic neurons [19]. While the CMA complex is formed by LAMP2A (lysosome-associated membrane protein type 2A), the chaperone protein Hsp70, and the substrate protein molecules that contain the five-peptide KFERQ-like structure, many other chaperones, such as Hsp40, EF1α, Hsp90, and Hip, interact with this trafficking complex [20,21]. There are ten human sHsps, of which hsp27 is the most abundant and ubiquitously expressed [22].

Both Parkinson’s disease and spastic paraplegia are protein-misfolding diseases, characterized by a reduction in DA due to loss of dopaminergic neurons and abnormal oxidation of DA [23]. 5-cysteinyl-dopamine (cys-DA) is among the products of DA oxidation present in both neuromelanin granules of substantia nigra and measurable in the cerebrospinal fluid (CSF) [24]. Hsp27 is also present in the substantia nigra and participates in the storage of polymerized DA oxidation products [25]. Here, we report a patient suffering spastic paraplegia with novel heterozygous mutations and analyzed the levels of hsp27, DA, and its metabolites 3,4-dihydroxyphenylacetic acid (DOPAC) and cys-DA in the CSF.

## 2. Methods

### 2.1. Ethics Approval and Informed Consent

The study project was approved by the Santa Lucia Hospital Clinical Ethics Committee (ref. CEI.22-18_SPG11s). Written informed consent was obtained from the patient’s legal ward to carry out the study with his biological samples and for the publication of any potentially identifiable images with scientific divulgation interest or data included in this article.

The patient completed large clinical examinations and tests from 2015 to 2020, such as determinations for toxic, metabolic, infectious, autoimmune, and inflammatory processes and had undergone genetic testing for common hereditary neurodegenerative disorders. The Spastic Paraplegia Rating Scale (SPRS) was registered upon admission (March 2020) and 1 year before [26]. Additional clinical details are summarized in Table 1.

### 2.2. Patient Sequencing

As previously described elsewhere by the authors [27], DNA was obtained and purified from the patient’s peripheral blood. We carried out whole-exome sequencing by using a QGenExWES kit (QGenomics, Barcelona, Spain) on a NovaSeq 6000 (Novogene, Cambridge, UK) at the Genetic Laboratory of Virgen Arrixaca University Hospital, Murcia, Spain. In accordance with the genome analysis best practices guidelines, we mapped the sequence reads to the human reference genome using the Burrows-Wheeler Aligner (v.0.8), removed duplicate reads (Picard v.1.11), and identified SNPs and indels (SAMtools v.1.0). Pathogenic and likely pathogenic variants considered to be of clinical significance were confirmed by Sanger sequencing. Unfortunately, his family declined testing for the segregation study.

### 2.3. Hsp27 Analysis in the CSF

Samples of cerebrospinal fluid were solved in lysis buffer. Protein concentrations were measured by using a bicinchoninic acid kit (Sigma-Aldrich, Hamburg, Germany). The hsp27 protein concentration was assayed by a hsp27 enzyme-linked immunosorbent assay (ELISA) kit (Thermo-Fisher Scientific, Waltham, MA, USA) according to the manufacturer’s instructions as previously published [28]. The detection range was 200–2200 pg/mL. Briefly, the human monoclonal antibody was first pre-coated in the culture plate. Approximately 50 µL of the kit standards or the test samples was added into the pit of the enzyme label-coated plate and then incubated at 37 °C for 30 min. An Hsp27-specific enzyme-conjugated antibody was added to the plate. Then, 50 µL of both A and B coloring reagents was added. The colored reaction intensity was determined at 450 nm with an automated ELISA reader. All samples had three independent replicates. The concentrations of the test samples were calculated according to the relationship between the absorbance values and the concentration test standards. Expression of the results was carried out in the form of pg/mL.

### 2.4. Measurement of Dopamine Levels in CSF

HPLC equipment, which included a dual pump and Nanospace SI-2/3001 pump (Shiseido Co., Tokyo, Japan), was used to assess the amounts of DA and its metabolites in the CSF samples. Dopamine and its metabolites standards were obtained from Sigma (Sigma-Aldrich, Hamburg, Germany) and from the Chemical Synthesis and Drug Supply Program of the National Institute of Mental Health. Previously, the CSF samples were centrifuged for 10 min at 4 °C with 12,000× *g*, filtered through a 0.20 μm membrane filter, and transferred to a high-performance liquid chromatography (HPLC) device. The detection limits for catechols were about 10 pmol/L or 10 fmol per assayed mL of CSF. The amount of DA in the CSF samples was measured as previously described [29]. Briefly, the injection volume was 10 µL into a 5 μm C18 reverse phase column. Standard solutions were prepared using the stock solutions of each standard. Approximately 1 mg of each standard was solved in 1 mL of 20% (*v*/*v*) acetonitrile/water to achieve a final concentration of 1000 μg/mL for the preparation of the internal standards. The CSF concentrations of biogenic amines and their metabolites were expressed as “pg/mg”.

### 2.5. Statistical Analysis

Data are expressed as the mean ± SEM. Statistical analysis was performed by using IBM SPSS 20.0 (SPSS Inc., Chicago, IL, USA) software with one-way ANOVA followed by the Bonferroni post hoc test and graphs designed by using GraphPad Prism 8.4.3 for Windows (GraphPad Software LLC, La Jolla, CA, USA). The results were considered significant when *p* < 0.05.

## 3. Results

A 49-year-old Mediterranean European Caucasian male with a history of epilepsy treated with valproate since he was 17, mild intellectual disability without any impact on his daily life, and bladder dysregulation was admitted to the neurology department in March 2020 for gait impairment and progressive cognitive decline in the last 5 years. His parents were 74 and 72 years old, both European Caucasian and healthy with no history of consanguinity or Parkinsonism. The patient has one sister and one brother (50 and 46 years old, respectively), both of whom are healthy. When registered, his newborn metabolic and hearing screening tests were normal. Speech–language and early psychomotor development, such as walking and sphincter control, were normally developed at the age of 1 and 2.5 years, respectively. He finished his primary studies and then lived with his parents and worked in a family business.

The patient was studied by the neurology department 1 year before he was admitted for progressive cognitive impairment and bradykinesia–rigidity syndrome. This was attributed to a long-term side effect of valproate. Nonetheless, no improvement was observed once valproate was suspended nor when the L-DOPA test was carried out. Suspecting atypical Parkinsonism, a Wilson disease study and genetic tests of Charcot–Marie–Tooth (CMT), atypical Parkinsonisms (PARKs, GAB, DNAJC6, VPS13C, PINK, and LG), and atypical spinal–muscular atrophies (DYNC1H1, BICD, VAPB, GARS, DYNC1H1, mtATP6, and mtATP8) were carried out. All of these genetic tests were normal.

Upon admission, neurological examination showed poor language ability, hypometric saccades in all directions, mild weakness in the right arm and in the left extremities, mainly in the foot, generalized hyperreflexia, non-stop ankle clonus, extensor plantar reflex, and generalized spasticity with impossibility to stand-up and walk. The Spastic Paraplegia Rating Scale (SPRS) scored 36 indicating severe disease. Brain MRI (Figure 1A–D) showed no alterations, including the thickness of the corpus callosum, while electromyography/electroneurography (EMG/ENG) and nerve conduction studies demonstrated mild and symmetric axonal sensorimotor polyneuropathy in the lower limbs but no signs of pathology of the anterior horn of the spinal cord. Therefore, suspecting atypical Parkinsonism, more genetic testing was ordered. Whole exome sequencing was performed in a genetic laboratory (Virgen Arrixaca Hospital, Murcia, Spain), and the obtained variants were prioritized based on known disease association and population frequency. Our patient had mutations in the SPG11 and SYNJ1 genes. The patient carried SPG-11 c.1951C>T and SYNJ1 c.2614G>T mutations, both heterozygous recessive, creating premature stop codons, p.R651* and p.D872Y*, respectively; producing aberrant spatacsin and synaptojanin-1 proteins. Unfortunately, the patient’s relatives refused to undergo genetic testing. Therefore, we were unable to acquire the family pedigree and assess if the patient’s mutations were inherited and/or spontaneous. Nowadays, there are only certain symptomatic treatments available for spastic paraplegia such as baclofen, botulinum toxin, and tizanidine used to ameliorate spasticity and urinary incontinence. Despite the patient being treated with those treatments and extensive physiotherapy, he needed a wheelchair. Future therapies should target restoring protein functions.

### DA, DA Oxidation, and Heat-Shock Protein 27

The SPECT DA Transporter (DAT) scan of the brain patient (Figure 1E,F,H) showed a significantly decreased optical density in both the left (41.80 ± 5.54, t_1,5_ = 9.802, *p* < 0.001) and right (77.95 ± 2.70, t_1,5_ = 6.890, *p* < 0.01) striates and putamen nuclei compared to a series of five cases in which the tests were considered normal by experienced radiologists (left: 98.94 ± 1.81%; right: 101.1 ± 1.99%). Accordingly, the patient’s DA and DOPAC levels in their CSF (Figure 1G,I) were significantly decreased (53.0 ± 6.92, t_1,3_ = 4.239, *p* < 0.05; 473.3 ± 72.19, t_1,3_ = 3.526, *p* < 0.05, respectively) compared to the healthy controls (91.33 ± 5.81; 916.7 ± 101.4, respectively). Furthermore, while no differences between the CSF samples were found in the Cys-DA levels (Figure 1J), the ratio Cys-DA/DOPAC (Figure 1K) was significantly higher in the patient’s CSF sample (0.213 ± 0.024, t_1,3_ = 2.788, *p* < 0.05) compared to the control samples (0.143 ± 0.007). Finally, the levels of hsp27 (Figure 1L) in the CSF were significantly higher in the patient (1073.0 ± 136.4, t_1,3_ = 3.532, *p* < 0.05) compared to the controls (589.2 ± 36.4).

## 4. Discussion

We found heterozygous recessive mutations in the SPG-11 (c.1951C>T) and SYNJ1 (c.2614G>T) genes. The present c.1951C>T SPG-11 mutation was previously described as a recessive compound heterozygous mutation in an isolated patient from Romania [30] and a Portuguese family [31]. In both reports, all patients had onset in the first decade of life and were clinically characterized by ataxia and cognitive regression. To the best of our knowledge, the c.2614G>T SYNJ1 mutation has not been previously reported. Although several mutations have been described in spastic paraplegia, about 50% of cases remain without a genetic diagnosis. There is a growing body of evidence suggesting that the wide genetic heterogeneity and the multiple functions of proteins related to spastic paraplegia indicate that the corticospinal tract is vulnerable to diverse metabolic and protein impairments. Therefore, spastic paraplegia may be caused by diverse molecular causes involved in common biochemical pathways. No cases of spastic paraplegia have been reported to be caused by monoallelic recessive SPG-11 or SYNJ1 mutations. Therefore, we suggest that the genetic interaction and protein haploinsufficiency of SPG11 and SYNJ1, both proteins implicated in common biochemical pathways, cause clinical spastic paraplegia of late onset. It is worth noting that the overlap of spastic-paraplegia-related genes with other neurodegenerative diseases may make the diagnosis of specific spastic paraplegia difficult. In this regard, patients diagnosed with mild-to-moderate SPG11 spastic paraplegia typically show minimal corpus callosum thinning while severe SPG11 patients had more severe thinning as well as cerebral atrophy [9]. We were not able to diagnose our patient with a known and well-described form of spastic paraplegia. Therefore, despite the clinical features being typical of spastic paraplegia, other atypical Parkinsonism should be considered.

Biochemically, spatacsin is a 280-kDa protein implicated in the axonal trafficking of vesicles and lysosomal homeostasis during autophagy [32] resulting in lipids and other neurotoxic proteins accumulation [33]. In addition, synaptojanin-1 is a 145-kDa essential inositol phosphatase enriched in the presynaptic terminal containing three functional domains involved in regulating synaptic vesicle recycling at various stages of synaptic activity [34]. Moreover, recent preclinical studies have shown that synj1 is required for autophagosome maturation at presynaptic DAergic terminals [35]. Thus, the patient does not produce fully functional spatacsin and synaptojanin-1 proteins. As previously reported, SYNJ1 haploinsufficiency leads to slowed synaptic vesicle endocytosis in midbrain neurons, α-synuclein accumulation, and neuron vulnerability [36]. In this line, our results showed decreased a DAT scan signal in the patient’s midbrain as well as decreased DA and DOPAC levels in their CSF, indicating the loss of DAergic neurons. Strikingly, while our results showed no differences in the Cys-DA levels, the ratio Cys-DA/DOPAC was significantly higher in the patient’s CSF sample. Given that DOPAC is the main intracellular DA metabolite, loss of nigrostriatal neurons may cause lower DOPAC levels in the CSF. Nonetheless, since cys-DA and DOPAC are both cytoplasmic, loss of DA neurons would be expected to produce proportionate decreases in both cys-DA and DOPAC and an unchanged ratio. Many factors have been described as being responsible for the increased cys-DA/DOPAC ratio [37]. It is worth noting that (1) MAO inhibition is known to increase intracellular DA and consequently increase its spontaneous oxidation. Despite MAO activity being reported to be normal in Parkinson’s disease [38], we cannot exclude this mechanism in our patient’s disease, and (2) decreased antioxidant activity in DAergic neurons would increase catechol quinones and consequently cysteinyl-catechols [39,40].

Finally, the CSF levels of hsp27 were enhanced in the patient. Small heat-shock proteins are well known because their cellular function is to stabilize and prevent protein misfolding and aggregation [41]. A recent study showed that even at low levels of DA_ox_, the hsp27 activity was significant in preventing amyloid fibrillar and amorphous aggregation of proteins [42]. Thus, we suggest that hsp27 is upregulated as a cellular mechanism of protection and/or to balance impaired protein trafficking. The underlying cellular and molecular mechanism of neurodegeneration in spastic paraplegia is not well known. It is theorized that neurodegeneration is caused by protein trafficking defects which leads to an imbalance in antioxidant proteins, DA vesicle trafficking/recycling processes, autophagy dysfunction, and cell death. Nonetheless, future studies are needed to understand and settle the role of hsp27 in spastic paraplegia. It will be necessary to improve the understanding of the pathophysiology of neurodegenerative diseases because it would facilitate the development of effective treatments to relieve or prevent neurodegeneration in patients with spastic paraplegia.

In conclusion, we report for the first time a known heterozygous recessive SPG11 c.1951C>T mutation and a not-reported heterozygous recessive SYNJ1 c.2614G>T mutation in a European Caucasian patient with clinical spastic paraplegia of late onset. We suggest that this complex genetic interaction causing haploinsufficiency of both proteins might be underlying the impairment in protein/vesicle intracellular trafficking and consequently the upregulation of hsp27. Finally, the diagnosis of spastic paraplegia should be considered in patients with unexplained neurodegeneration especially when accompanied by a history of atypical Parkinsonism.

## Figures and Tables

**Figure 1 genes-14-01320-f001:**
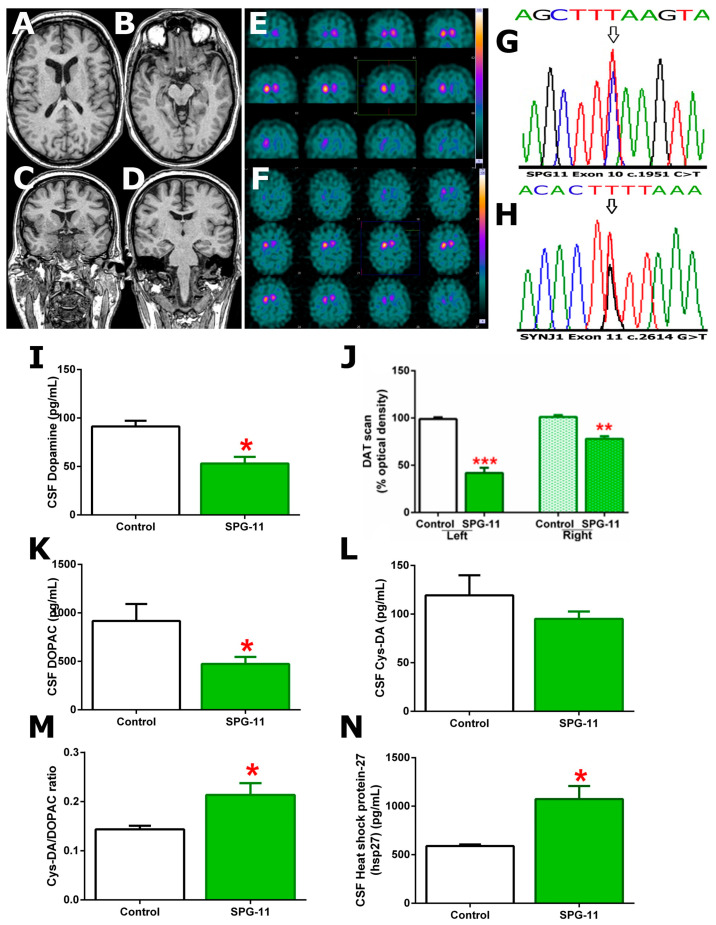
Clinical and functional impact of heterozygous recessive SPG11 and SYNJ1 mutations. Patient’s brain MRI (**A**–**D**) showing no anatomical alterations. Patient’s DAT-scan images (**E**,**F**) showing non-symmetric decreased optical density in both basal ganglia. Sanger chromatogram (**G**,**H**) demonstrating relevant mutations in the SPG11 and SYNJ1 alleles from the patient (arrows). Decreased DA levels (**I**) in the CSF samples from the patient and the controls (* *p* < 0.05). DAT scan optical density analysis (**J**) showing decreased intensity in both, left and right, basal ganglia from the patient compared to the healthy controls (** *p* < 0.01 and *** *p* < 0.001). DA oxidation metabolites (**K**–**M**) in the CSF from the patient showing decreased DOPAC levels (* *p* < 0.05) compared to the controls; no differences in cys-DA levels but increased cys-DA/DOPAC ratio (* *p* < 0.05) compared to the controls. The patient’s hsp27 CSF levels (**N**) were increased (* *p* < 0.05) compared to the controls.

**Table 1 genes-14-01320-t001:** Spastic paraplegia patient’s molecular, clinical and neurological findings.

Mutations	SPG11 c.1951C>T, p.R651* (Ch. 15, exon 10)
	SYNJ1 c.2614G>T, p.D872Y (Ch. 21, exon 11)
Age	48
Ethnicity	Caucasian, Mediterranean European
Symptom prompting first neurological examination	Cognitive decline at 47 years of age
Sequence of neurological symptoms	1 epilepsy crisis, cognitive decline, spasticity, gait impairment
Height (cm); weight (Kg)	175 cm; 83 Kg; BMI = 27
Developmental milestones	Normal
Education	Primary school
Activities of daily living	Independent until gait impairment
Resting tremor	No
Slurred speech	No
Tendon reflexes; plantar response	Hyperreflexia with clonus; extensor
Levodopa responsive	No
Urinary incontinence	No
Brain MR imaging	Normal
DAT-scan	DAT loss in *substantia nigra*
EMG/ENG	Sensorimotor polyneuropathy in lower limbs
SPRS on admission	36/52
SPRS 1 year before	9/52

Abbreviations: Ch. = chromosome; BMI = body mass index; SPRS = spastic paraplegia rating score; DAT = dopamine transporter; EMG/ENG = electromiography/electroneurography.

## Data Availability

The datasets for this article are not publicly available due to concerns regarding participant/patient anonymity. Requests to access the datasets should be directed to the corresponding author.

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
