# Peer review of "Upregulation of Heat-Shock Protein (hsp)-27 in a Patient with Heterozygous SPG11 c.1951C>T and SYNJ1 c.2614G>T Mutations Causing Clinical Spastic Paraplegia"

_genes, 2023, doi:10.3390/genes14071320_

Round 1

Reviewer 1 Report

This study reports a case of a 49-year-old patient with spastic paraplegia and a novel heterozygous mutation. The whole exome sequencing revealed a SPG11 c.1951C>T mutation and a SYNJ1 c.2614G>T mutation, both are heterozygous recessive and create premature stop codons. Through MRI and cerebrospinal fluid (CSF) analyses, the authors found that the patient had dopamine transporter (DAT) loss in the substantia nigra, decreased CSF DA and DOPAC levels, and increased CSF HSP27 levels. The authors conclude that haploinsufficiency of SPG11 and SYNJ1 might contribute to the neurodegeneration, probably via impaired protein trafficking. This is a valuable case study. I have some comments that may improve the manuscript:

1. Authors shall notice that mutations in the SYNJ1 gene are associated with early-onset Parkinson's disease (OMIM #615530). Poor or no response to levodopa is a common feature to atypical parkinsonism. In addition, the DAT-scan showed DAT loss in the substantia nigra. Therefore, atypical Parkinsonism can not be excluded. In my opinion, the heterozygous SYNJ1 c.2614G>T mutation may not cause early-onset but may delay the onset of Parkinson’s disease, since it shall be a severe loss of function allele. This shall be carefully discussed.

2. Is the SYNJ1 mutation inherited paternally or maternally? Does the family with this mutation have a history of late-onset Parkinsonism?

3. In the SPG11 associated spastic paraplegia patients, the MRI data often show abnormal white matter changes, such as thin corpus callosum (OMIM #604360). Did the patient have serial MRIs that show any progressive atrophy of the white matter (eg, the corpus callosum)? This may strengthen the diagnosis.  

4. Some abbreviations do not have full names, eg: EMG, ENG.

The language is understandable, but can be improved. 

Author Response

Reviewer 1

This study reports a case of a 49-year-old patient with spastic paraplegia and a novel heterozygous mutation. The whole exome sequencing revealed a SPG11 c.1951C>T mutation and a SYNJ1 c.2614G>T mutation, both are heterozygous recessive and create premature stop codons. Through MRI and cerebrospinal fluid (CSF) analyses, the authors found that the patient had dopamine transporter (DAT) loss in the substantia nigra, decreased CSF DA and DOPAC levels, and increased CSF HSP27 levels. The authors conclude that haploinsufficiency of SPG11 and SYNJ1 might contribute to the neurodegeneration, probably via impaired protein trafficking. This is a valuable case study. I have some comments that may improve the manuscript:

  1. Authors shall notice that mutations in the SYNJ1 gene are associated with early-onset Parkinson's disease (OMIM #615530). Poor or no response to levodopa is a common feature to atypical parkinsonism. In addition, the DAT-scan showed DAT loss in the substantia nigra. Therefore, atypical Parkinsonism can not be excluded. In my opinion, the heterozygous SYNJ1 c.2614G>T mutation may not cause early-onset but may delay the onset of Parkinson’s disease, since it shall be a severe loss of function allele. This shall be carefully discussed. 

We thank the reviewer for this suggestion. We agree and have clarified it in the text including the following paragraph and new references. 

“Although several mutations have been described in spastic paraplegias about 50% of cases remains without a genetic diagnosis. There is a growing body of evidence suggesting that the wide genetic heterogeneity and the multiple functions of proteins related to spastic paraplegias indicate that corticospinal tracts are vulnerable to a diverse metabolic and protein impairments. Therefore, spastic paraplegias may be caused by diverse molecular causes involved in common biochemical pathways. No cases of spastic paraplegia have been reported caused by  monoalellic recessive SPG-11 or SYNJ1 mutations. Therefore, we suggest that the genetic interaction and protein haploinsufficiency of SPG11 and SYNJ1, both proteins implicated in common biochemical pathways, are causing clinical spastic paraplegia of late onset. It is worthy to note that the overlap of spastic paraplegia related genes with other neurodegenerative diseases may difficult the diagnosis of specific spastic paraplegias. In this regard, patients diagnosed with mild-to-moderate SPG11 spastic paraplegia typically shown minimal corpus callosum thinning while severe SPG11 patients had more severe thinning as well as cerebral atrophy [9]. We were not able to diagnose our patient with a known and well described spastic paraplegia. Therefore, despite clinical features being typical of spastic paraplegia other atypical parkinsonism should be considered.”

  1. Is the SYNJ1 mutation inherited paternally or maternally? Does the family with this mutation have a history of late-onset Parkinsonism?

We thank the reviewer for this question. That’s a good one. Unfortunately, his parents, sister and brother declined genetic testing. All of them are healthy without signs or history of parkinsonism. In the present version of the manuscript we have clarified this point in the results as follows: 

…”His parents were 74 and 72 years old, both European Caucasian and healthy with no history of consanguinity or parkinsonism. The patient had 1 sister and 1 brother (50 and 46 years old, respectively) both healthy”…

...”We suggested genetic testing for the patient's relatives however they declined. Therefore, we are unable to assess if the mutations were inherited and/or spontaneous.”

  1. In the SPG11 associated spastic paraplegia patients, the MRI data often show abnormal white matter changes, such as thin corpus callosum (OMIM #604360). Did the patient have serial MRIs that show any progressive atrophy of the white matter (eg, the corpus callosum)? This may strengthen the diagnosis.  

We thank the reviewer for this key question. We diagnosed our patient with spastic paraplegia but not specifically with SPG11 spastic paraplegia. 

We measured by an experienced neuroradiologist the thickness of corpus callosum among other structures looking for these features. Nonetheless, in our patient the thickness is normal. In the present version of the manuscript we included a paragraph about this in the case presentation and in the discussion sections.

In this regard, last evidence (Kara et al., 2016) showed that patients with mild-to-moderate SPG11 disease had minimal corpus callosum thinning while severe SPG11 patients had more severe thinning as well as cerebral atrophy. When the disease is severe the corpus callosum remains at a static state and does not seem to change over time. We have a 2-years post control MRI without changes. We've planned future MRI studies at 5  (2025) and 10 years (2030).

Ref: Kara E, Tucci A, Manzoni C, Lynch DS, Elpidorou M, Bettencourt C, Chelban V, Manole A, Hamed SA, Haridy NA, Federoff M, Preza E, Hughes D, Pittman A, Jaunmuktane Z, Brandner S, Xiromerisiou G, Wiethoff S, Schottlaender L, Proukakis C, Morris H, Warner T, Bhatia KP, Korlipara LV, Singleton AB, Hardy J, Wood NW, Lewis PA, Houlden H. Genetic and phenotypic characterization of complex hereditary spastic paraplegia. Brain. 2016 Jul;139(Pt 7):1904-18. doi: 10.1093/brain/aww111. Epub 2016 May 23. PMID: 27217339; PMCID: PMC4939695.

  1. Some abbreviations do not have full names, eg: EMG, ENG.

We thank the reviewer for catching it. We have corrected it. 

Reviewer 2 Report

In this manuscript, Juan Antonio García-Carmona et al reported heterozygous recessive mutations of SPG-11 and SYNJ1 genes in one clinical spastic paraplegia case. ELISA and HPLC study of patient CSF found DA and DOPAC levels in CSF were significantly decreased, while no differences were found in Cys-DA; Cys-20 DA/DOPAC ratio and hsp27 levels were significantly higher. This is an interesting study with the finding of novel mutation in SYNJ1, which might expand the causative genes for spastic paraplegia. I have a few small concerns here.

1. Whole exome sequencing is performed, but sanger sequencing is preferred to double confirm the genetic mutations.

2. In the first paragraph of Discussion - "SPG-11 mutation was previously described as recessive compound heterozygous mutation in an isolated patient from Romania (Stevanin et al., 2008) and a Portuguese family (Santos et al., 2022). In both reports, all patients had onset in the first decade of life and clinically were characterized by ataxia, cognitive regression. For this 49-year-old Caucasian male patient, mild intellectual disability was observed only after he was 17. Therefore, we suggest that the haploinsufficiency of both proteins are causing clinical spastic paraplegia of late onset"  Does the author suggest that SPG-11 mutation alone cause early onset of spatic paraplegia, while SPG-11 and  SYNJ1 mutations cause late onset of symptom? What is the interactions of these two mutations and their contribution to the clinical symptoms?

3. In the first paragraph of Discussion, "monoalelic" should be "monoallelic".  

Author Response

Reviewer 2

In this manuscript, Juan Antonio García-Carmona et al reported heterozygous recessive mutations of SPG-11 and SYNJ1 genes in one clinical spastic paraplegia case. ELISA and HPLC study of patient CSF found DA and DOPAC levels in CSF were significantly decreased, while no differences were found in Cys-DA; Cys-20 DA/DOPAC ratio and hsp27 levels were significantly higher. This is an interesting study with the finding of novel mutation in SYNJ1, which might expand the causative genes for spastic paraplegia. I have a few small concerns here.

  1. Whole exome sequencing is performed, but sanger sequencing is preferred to double confirm the genetic mutations.

We thank the reviewer. In the present version of the manuscript we have included the sanger sequencing in the text and in the figure 1. 

  1. In the first paragraph of Discussion - "SPG-11 mutation was previously described as recessive compound heterozygous mutation in an isolated patient from Romania (Stevanin et al., 2008) and a Portuguese family (Santos et al., 2022). In both reports, all patients had onset in the first decade of life and clinically were characterized by ataxia, cognitive regression. For this 49-year-old Caucasian male patient, mild intellectual disability was observed only after he was 17. Therefore, we suggest that the haploinsufficiency of both proteins are causing clinical spastic paraplegia of late onset"  Does the author suggest that SPG-11 mutation alone cause early onset of spastic paraplegia, while SPG-11 and  SYNJ1 mutations cause late onset of symptom? What are the interactions of these two mutations and their contribution to the clinical symptoms?

We thank the reviewer for this question. SPG11 mutations have been described causing both, early and late onset spastic paraplegias. We were not able to diaWe have clarified that we suggest that the interaction between both SPG11 and SYNJ1 haploinsufficiency are causing the spastic paraplegia of the patient. We have included the following paragraph in the discussion section: 

“Although several mutations have been described in spastic paraplegias about 50% of cases remains without a genetic diagnosis. There is a growing body of evidence suggesting that the wide genetic heterogeneity and the multiple functions of proteins related to spastic paraplegias indicate that corticospinal tracts are vulnerable to a diverse metabolic and protein impairments. Therefore, spastic paraplegias may be caused by diverse molecular causes involved in common biochemical pathways. No cases of spastic paraplegia have been reported caused by  monoalellic recessive SPG-11 or SYNJ1 mutations. Therefore, we suggest that the genetic interaction and protein haploinsufficiency of SPG11 and SYNJ1, both proteins implicated in common biochemical pathways, are causing clinical spastic paraplegia of late onset”

… “We were not able to diagnose our patient with a known and well described spastic paraplegia. Therefore, despite clinical features being typical of spastic paraplegia other atypical parkinsonism should be considered.”

  1. In the first paragraph of Discussion, "monoalelic" should be "monoallelic". 

We thank the reviewer for catching it. We have corrected it. 

Round 2

Reviewer 1 Report

The authors have carefully addressed my comments. I don't have further concerns.